# Bioelectrical Impedance Vector Analysis, Nutritional Ultrasound^®^, and Handgrip Strength as Innovative Methods for Monitoring Critical Anorexia Nervosa Physical Recovery: A Pilot Study

**DOI:** 10.3390/nu16101539

**Published:** 2024-05-20

**Authors:** Jose M. Romero-Márquez, María Novo-Rodríguez, Cristina Novo-Rodríguez, Víctor Siles-Guerrero, Isabel Herrera-Montes, Francisco Garzón Navarro-Pelayo, Martín López-de-la-Torre-Casares, Araceli Muñoz-Garach

**Affiliations:** 1Department of Endocrinology and Nutrition, Virgen de las Nieves University Hospital, 18014 Granada, Spain; marianovor@correo.ugr.es (M.N.-R.); cristina.novo.sspa@juntadeandalucia.es (C.N.-R.); victor.siles.sspa@juntadeandalucia.es (V.S.-G.); isabel.herrera.montes.sspa@juntadeandalucia.es (I.H.-M.); martin.lopeztorre.sspa@juntadeandalucia.es (M.L.-d.-l.-T.-C.); 2Foundation for Biosanitary Research of Eastern Andalusia—Alejandro Otero (FIBAO), 18012 Granada, Spain; 3Granada Biosanitary Research Institute (Ibs. Granada), 18014 Granada, Spain; 4Eating Disorders Hospital Unit, Virgen de las Nieves University Hospital, 18014 Granada, Spain; francisco.garzon.n.sspa@juntadeandalucia.com; 5Network Biomedical Research Center Physiopathology of Obesity and Nutrition (CiberOBN), Carlos III Health Institute, 28029 Madrid, Spain

**Keywords:** anorexia nervosa, body composition, anthropometric, BIVA, HGS, dynamometer, muscle mass, fat mass, rectus femoris, adipose tissue, nutrition

## Abstract

Eating disorders (EDs) manifest as persistent disruptions in eating habits or related behaviors, significantly impacting physical health and psychosocial well-being. Nutritional assessment in ED patients is crucial for monitoring treatment efficacy. While dual-energy X-ray absorptiometry (DEXA) remains standard, interest in alternative methods such as bioelectrical impedance vector analysis (BIVA) and Nutritional Ultrasound^®^ (NU) has risen due to their affordability and portability. Additionally, hand dynamometry offers a user-friendly approach to assessing grip strength (HGS), indicative of nutritional status. A prospective study was carried out to evaluate the utility of BIVA, NU^®^, and HGS in 43 female AN patients. Measurements were taken at baseline and hospital discharge. A total of 41 patients completed the study. After the intervention, numerous BIVA-related parameters such as fat (3.5 ± 2 kg vs. 5.3 ± 2.7 kg, *p* < 0.001) and free fat mass (33.9 ± 3.8 kg vs. 37.5 ± 4.1 kg, *p* < 0.001) were partially restored. Similarly, Nutritional Ultrasound^®^ showed promising results in assessing body composition changes such as total abdominal fat tissue (0.5 ± 0.3 cm vs. 0.9 ± 0.3 cm, *p* < 0.05). In the same way, rectus femoris cross-sectional area values correlated with clinical outcomes such as free fat mass (0.883, *p* < 0.05) and appendicular muscle mass (0.965, *p* < 0.001). HGS reached the normality percentile after the intervention (21.6 ± 9.1 kg vs. 25.9 ± 12.3 kg, *p* < 0.05), demonstrating a significant association between grip strength and body composition parameters such as free fat mass (0.658, *p* < 0.001) and appendicular muscle mass (0.482, *p* < 0.001). Incorporating BIVA-, NU^®^-, and HGS-enhanced nutritional assessment into the treatment of AN patients offers cost-effective, portable, and non-invasive alternatives to DEXA. These techniques offer valuable insights into changes in body composition and nutritional status, which, in turn, facilitate treatment monitoring and contribute to improved patient outcomes.

## 1. Introduction

EDs manifest as persistent disruptions in eating habits or related behaviors, significantly impacting physical health and psychosocial well-being. EDs typically emerge during adolescence, carrying significant implications for both physical and mental well-being. Addressing these disorders involves a multifaceted therapeutic approach, necessitating the involvement of various medical specialties [1]. Among these disorders, anorexia nervosa (AN) is characterized by severe dietary restriction leading to a dangerously low body weight, driven by an intense fear of weight gain and a distorted body image [2]. AN predominantly affects girls and young women, with the highest risk occurring between ages 10 and 24. The incidence and prevalence of AN have surged within this demographic, particularly since the onset of the COVID-19 pandemic [3]. Surprisingly, a recent cross-sectional study conducted on 730 adolescents from Murcia (Spain) demonstrated that 30% of the adolescents presented with disordered eating patterns, which were associated with female sex, immigrant status, and excess weight [3]. Despite concerted therapeutic efforts, treatment efficacy remains modest, with remission rates fluctuating between 40% and 60% for AN and eating disorders not otherwise specified [1,4]. This variance in remission rates is partly ascribed to the heterogeneous definition of remission, which should encompass psychological, cognitive, behavioral (such as binge eating episodes or purging behaviors), and physical aspects (classically, body mass index [BMI]) [5]. Furthermore, relapses are frequent, particularly post-hospital discharge, underscoring the importance of suitable follow-up strategies. Therefore, international guidelines [6] recommend both psychological and physical interventions for monitoring the effectiveness of the treatment for individuals with AN. Historically, anthropometric measurements have served as the primary method for assessing nutritional status and body composition in AN patients. However, these measurements (such as BMI or only weight) may not adequately differentiate between key body compartments, reflecting methodological limitations [2]. In fact, in a meta-analysis with AN patients, the primary outcome considered was solely body weight. The study revealed that adolescents experienced faster weight gain compared to adults, but this was not associated with psychological findings in treating adults with AN [7].

Currently, DEXA, magnetic resonance imaging (MRI), and computed tomography (CT) are considered the gold standard techniques for body composition analysis [8]. However, numerous constraints continue to impede their widespread adoption in routine practice. Firstly, these techniques incur significant costs and demand skilled professionals for their administration and interpretation, often requiring specialized post-processing procedures [9]. Additional challenges include patient compliance issues, such as the hyperactivity frequently observed in AN patients [10], potentially compromising the quality of scanned images and subsequent analysis. Moreover, these patients may undergo multiple evaluations, leading to heightened exposure to ionizing radiation owing to the increased radiation doses associated with these imaging modalities.

These limitations have sparked interest in alternative methods such as BIVA. BIVA offers advantages such as affordability, portability, speed, and the absence of radiation exposure, analyzing impedance vectors and phase angle data to assess body water distribution, body cell mass, and cellular integrity, serving as indicators of nutritional status [11]. Similarly, NU^®^ employs ultrasound technology to target fat-free mass and fat mass, presenting an emerging, cost-effective, portable, and non-invasive solution. With linear, broadband, multifrequency probes capable of assessing the musculoskeletal area in-depth, it quantifies muscle modifications associated with malnutrition, providing valuable insights into functional changes within the body [12]. Additionally, the hand dynamometer provides a quick, user-friendly, and cost-effective method for assessing grip strength and, consequently, nutritional status. In fact, clinical studies across various patient populations have linked reduced grip strength, measured by hand dynamometry, with prolonged hospital stays, higher mortality rates, and increased complications [13].

The present research integrates the three aforementioned methods to conduct an in-depth characterization of body composition, specifically targeting muscle and body fat composition, as well as muscle function, in hospitalized AN patients. This research supplements these methods with laboratory parameters to elucidate the relationships among them. Therefore, the hypothesis of the present study is that the combined use of these methods will enable comprehensive monitoring of weight homeostasis recovery and enhance follow-up strategies for assessing the physical status of AN patients.

## 2. Materials and Methods

### 2.1. Study Design

This clinical practice study included 43 patients with a mean age of 28.7 ± 13.5 years who had been admitted to the Eating Disorders Hospitalization Unit (EDHU) of Virgen de las Nieves University Hospital from 2020 to 2023. Prior to admission, all female patients were diagnosed with anorexia nervosa (41) or eating disorders not otherwise specified (2) (EDNOS) according to DSM-V [14]. The inclusion criteria were the following: aged 16 years or older, a BMI of 14 or below, demonstrating genuine motivation for change and awareness of their illness, a confirmed diagnosis of AN, bulimia nervosa, or EDNOS with the severity not classified as mild, exhibiting a negative response to outpatient treatment, experiencing overwhelming or conflictive family dynamics, and displaying a tendency towards social isolation stemming from the illness.

### 2.2. Psychiatric and Nutritional Intervention in EDHU

Some outcomes were monitored during the EDHU hospitalization program, including the normalization of eating patterns, food exposure, intervention on compensatory behaviors such as compulsive physical exercise, vomiting, or the use of laxatives, acquisition and improvement of disease awareness, and restructuring of the main beliefs, thoughts, and attitudes, as well as basic altered emotions, about diet, weight, and body image. The psychiatric and nutritional care comprised a therapeutic dining room to restore eating patterns, with the goal of recovering from physical and environmental problems. Similarly, an eating behaviors intervention was implemented with the goal of normalizing eating behavior and aiding the transition to an outpatient setting. The idea was to offer tailored attention to each disorder’s most defining eating patterns in the present moment. A thorough inspection of the tray prepared for the event was conducted, including all of the things previously established on the menu (sugar, oil, etc.). A registered dietitian prescribed a diet based on each patient’s calorie and protein requirements. The diet was validated by medical indication (psychiatry and endocrinology units) and monitored by nursing personnel. Additionally, the EDHU presents a reliable protocol to prevent refeeding syndrome, which involves the administration of vitamins B1, B6, and B12, along with serum therapy recommendations. The decision to include or exclude potassium chloride, monosodium phosphate, and magnesium sulfate in serum therapy was determined by analytical findings.

### 2.3. Anthropometric Measurements

At baseline, a stadiometer was used to measure height, and weight was calculated using a calibrated weighing scale set (certified test weights ± 0.1 kg) (SECA 665, Hamburg, Germany). Calf (CC) and arm circumferences (AC), as well as triceps skinfold thickness (TST), were measured according to recommendations [15]. All measurements were taken during hospital admission and release. The procedure, performed by experienced professionals, aimed to minimize measurement variability at hospital admission and discharge.

### 2.4. Bioelectrical Impedance Vector Analysis

Whole-body BIVA measurements were conducted using a 50 kHz phase-sensitive impedance analyzer (BIA 101 AKERN, Pontassieve, Italy) with tetrapolar 800 mA wearable electrodes on the right hand and foot as previously reported [16]. The body’s complex circuits, involving resistance (Rz) and reactance (Xc) elements, were stimulated with an alternating current to determine phase angle (PhA). According to standard protocol [17], one impedance adhesive electrode (Biatrodes Akern Srl, Florence, Italy) was placed on the back of the right hand (center of the third proximal phalanx) and the other electrode on the neck of the corresponding foot (proximal to the second and third metatarsophalangeal joints). BIVA interpretation, introduced by Piccoli et al. [18], involves plotting standardized Rz and Xc values on a resistance—reactance graph, enabling direct assessment of impedance without relying on body weight, equations, or models. Bioelectrical parameters were analyzed to estimate body composition, including fat mass (FM), fat-free mass (FFM), body cell mass (BMC), total muscle mass (TMM), appendicular skeletal muscle mass (ASMM), total body water (TBW), and extracellular body water (ECW). The procedure, performed by experienced professionals, aimed to minimize measurement variability at hospital admission and discharge.

### 2.5. Nutritional Ultrasound^®^

NU was carried out as previously reported [16]. Briefly, a HITACHI ALOKA F37 ultrasound scanner (Hitachi healthcare, Tokyo, Japan) and an Aloka UST-5413 Linear Array (10–12 MHz) transducer (Hitachi healthcare, Tokyo, Japan) were employed. Patients were positioned supine with a specified limb alignment and assessed by experienced specialists using water-soluble transmission gel. The pictures of the right rectus femoris (RF) muscle were taken one-third of the way between the patella and the iliac crest, beginning at the patella. The rectus femoris cross-sectional area (RF-CSA), RF axis (-X and -Y axes), and leg subcutaneous fat (L-SAT) were all measured as presented in Figure 1A. Rectus femoris was selected due to the correlation with metabolically active FFM [12]. The examination extended at the abdominal level, measuring at the midway between the xiphoid appendix and the navel. Measures comprised total subcutaneous abdominal adipose tissue (T-SAT), superficial subcutaneous abdominal fat (S-SAT), and total visceral adipose tissue (VAT), which were associated with the amount of fat deposits and their distribution (Figure 1B) [12]. The procedure, performed by experienced professionals, aimed to minimize measurement variability at hospital admission and discharge.

### 2.6. Handgrip Strength Analysis

The measurement of hand grip strength (HGS) in the dominant hand was performed using a Jamar dynamometer (Asimow Engineering Co., Los Angeles, CA, USA). Patients were positioned in a seated posture with the wrist and forearm in a neutral position, the elbow bent at a 90-degree angle, the forearm neutrally rotated, and the shoulder adducted. To ensure accuracy, the mean value was calculated by instructing patients to perform three consecutive contractions spaced one minute apart as previously reported [16]. The procedure, performed by experienced professionals, aimed to minimize measurement variability at hospital admission and discharge.

### 2.7. Biochemical Analysis

Biomolecular markers were assessed to analyze nutrition and inflammation status, including glucose, creatinine, proteins, albumin, prealbumin, phosphorus, calcium, magnesium, potassium, C-reactive protein (CRP), total cholesterol (TC), and triglycerides levels.

### 2.8. Statistical Analysis

The statistical program IBM SPSS 25 (Chicago, IL, USA) was used to examine normality, variance homogeneity, and Pearson’s correlation. Physical recovery results were analyzed using paired sample *t*-tests, with *p*-values < 0.05 indicating significance. In addition, the MetaboAnalyst V5.0 software was used to perform Partial Least Squares Discriminant Analysis (PLS-DA). PLS-DA was used to assess the normalized and auto-scaled mean values of the 32 variables collected from the patients. In this PLS-DA, the variables of significance in projection (VIP) score selection criteria were values greater than one, equivalent to *p* < 0.05. The current study used an intention-to-treat analysis, with two patients who did not complete therapy included in the first analysis.

## 3. Results and Discussion

### 3.1. Body Composition Analysis in Critical AN Patients

A total of 41 patients completed the study, spending a mean of 49 ± 20.2 days in the unit. The patients had a mean age of 28 ± 13.2 years. Table 1 details the population characteristics upon hospital admission and discharge. As anticipated, the nutritional and psychiatric interventions led to an increase in body weight and BMI, aligning with the standard physical monitoring protocol for AN patients, as outlined previously [2,19]. Interestingly, the AC values increased after the intervention. This finding is corroborated by the TST value, which was higher at discharge, suggesting increased body fat deposition in the upper extremities. Although not statistically significant, there was a noticeable trend towards an increase in CC, suggesting a potential recovery of tissue in this region.

According to BIVA parameters, the PhA values did not change after the intervention. However, three studies have suggested that PhA improved following nutritional intervention in AN patients. Interestingly, these studies, which assessed the physical recovery of AN patients, reported significantly higher body weight and BMI compared to those presented in this study [2,19,20]. Indeed, the presented PhA was lower than that reported in the scientific literature [2,19], likely attributable to the critical condition of AN patients upon hospital admission. Notably, in 2012, Haas et al. demonstrated that short-term multidisciplinary interventions did not alter PhA in AN patients with comparable body weight and BMI to those in this study [21]. These results suggest that in cases of extremely low body weight, PhA might not serve as a reliable predictive tool for physical recovery in critical AN patients. In contrast, the FM and FFM values increased after the intervention, aligning with the reported body weight gain [2,19,20]. Likewise, the intervention resulted in increased ASMM values, as corroborated by the AC and CC values. This increase in muscle mass was accompanied by rises in extracellular and total body water content, consistent with findings from previously published studies that monitored changes in body composition during refeeding of patients with AN [2,19,22,23]. In recent years, there has been a growing focus on muscle gain as part of AN recovery efforts. Indeed, a recent systematic review demonstrated that therapeutic exercise led to an increase in muscle mass and was associated with improvements in anorexia symptoms, as well as physical and mental health [24]. Consequently, prioritizing muscle regain should be a central aspect of physical recovery for these patients.

The ultrasound adipose-related parameters indicated an increase in T-SAT values, mirroring the observed increase in FM and supporting adipose tissue gain, particularly in the trunk. These results were supported by Lackner et al., who showed similar upper abdominal SAT values in AN patients using a linear probe (L8-18i RS) with similar conditions (8–16 MHz) [25]. Similarly, although not statistically significant, a slight trend was observed in S-SAT and L-SAT values. These findings align with studies that assessed body fat distribution using the DEXA method in AN patients. These studies demonstrated that following partial weight restoration, body fat deposition was more pronounced in the trunk region compared to the legs [26,27,28].

In contrast, the ultrasound muscle-related parameter did not detect slight variations in muscle gain. This could be attributed to the small variation in ASMM gain during the stay, which may not have been sufficient to manifest in the spatial muscle distribution of RF. These results are consistent with findings by Franzoni et al., who evaluated muscle content in the total body and lumbar spine (L1–L4) using the DEXA method in AN patients. Following a 12-month multidisciplinary intervention, an increase in body weight was observed but was not associated with local or total skeletal muscle mass gain [26]. Indeed, a systematic review investigating muscle recovery in AN patients post-intervention suggests lower muscle mass despite weight regain in this population. While these differences often did not reach significance in individual studies, the general trend in the current literature points towards incomplete muscle recovery after AN [29]. These findings are particularly compelling due to the high replicability and correlation observed between NU and DEXA methods. The ultrasound approach is a swift and reliable procedure that enables the direct evaluation of muscular and adipose tissue distribution during medical consultations. Therefore, NU could serve as a valuable tool for assessing the location-specific distribution of adipose and muscular tissue, potentially reducing healthcare costs and time spent on body composition evaluation for these patients.

According to body strength, limited evidence suggests that AN patients exhibit lower strength in both their legs [30,31] and arms [30] compared to healthy BMI-matched individuals in these studies. In this study, the mean initial HGS values were 21.6 ± 9.1 kg, which corresponds to the 25th percentile of the Andalusian population as reported using a validated Jamar dynamometer [13]. To the best of our knowledge, this is the first study to utilize HGS for monitoring physical recovery in AN patients. Interestingly, following multidisciplinary intervention, AN patients demonstrated an increase in mean HGS values (25.9 ± 12.3 kg), reaching the 50th percentile [13]. The hand dynamometer provides a swift, easy-to-use, and economical approach with which to evaluate grip strength, which can indicate the nutritional health of patients. Many clinical investigations involving various patient groups (such as those undergoing surgery, elderly individuals, cancer patients, etc.) have demonstrated that lower grip strength, assessed via hand dynamometry, is linked to longer hospital stays, elevated mortality rates, and heightened complications [13]. Therefore, potential rehabilitation programs or interventions should include physical activity, focused on strengthening muscle mass.

Most of the AN patients were admitted to the EDHU in a hypoglycemic state, which is a common feature in this population [32,33]. Interestingly, the multidisciplinary intervention successfully restored blood glucose levels to normal values, thereby reducing the potential occurrence of associated clinical comorbidities. In contrast, markers related to protein metabolism (creatinine, proteins, albumin, and prealbumin) were not affected by the multidisciplinary intervention. It is worth noting that all parameters remained within the normal range according to the Andalucía Health System (AHS). These findings were supported by a systematic review conducted by Lee et al., which evaluated the modulation of serum protein levels in both pathological and non-calorically restricted patients [34]. In this study, the albumin and prealbumin ranges in AN patients, matched for BMI, were similar to those presented in this manuscript, with modifications only observed in cases of extremely low BMI (<10 kg/m^2^) [34]. These findings suggest that in such cases, serum protein markers may not be optimal for monitoring physical recovery in AN patients.

On the other hand, lipid-related metabolism showed a reduction in triglyceride levels but not in TC levels after weight gain. A recent systematic review with meta-analysis revealed a lack of consistency in results regarding these parameters following partial weight restoration. The authors noted that serum lipid modification varies significantly depending on the initial status of AN patients, indicating the need for more evidence in this field to better understand the pathophysiology of AN [35].

CRP is an acute-phase protein commonly used in biochemical analysis to monitor inflammation or infectious processes [36]. In the case of AN patients, four cross-sectional studies [37,38,39,40] and one longitudinal study [41] have indicated that AN patients typically exhibit lower CRP serum levels compared to BMI-matched healthy controls. In this study, the mean CRP value of the AN patients before the multidisciplinary intervention fell within the range defined by the AHS, with some values skewing towards the upper limit, indicating a trend towards being elevated. Interestingly, after the intervention, the serum CRP decreased by 8.6 times, with minimal dispersion of the data. The only study that evaluated changes in serum CRP levels in AN patients demonstrated no significant difference in CRP levels before and after weight gain. Remarkably, the initial BMI (16.7 ± 1.2 kg/m^2^) and serum CRP levels (0.32 ± 0.25 mg/L) were similar to those obtained after the intervention in this research [41]. Considering the critical condition of the presented AN patients, these results suggest that after partial weight gain, CRP levels might be partially restored and may not be sensitive to slight weight gain, as reported in other studies [41]. Interestingly, an elevation in inflammatory markers has been linked to appetite suppression [42]. While there is no direct evidence of the correlation between CRP levels and appetite reduction, lowering CRP levels may represent a significant goal to pursue during the physical recovery phase of AN.

Finally, phosphorus but not calcium, magnesium, or potassium serum levels were increased after the intervention. Maintaining phosphorus homeostasis is crucial in managing AN due to refeeding hypophosphatemia (RH), which is commonly observed in these patients and complicates treatment [43]. These findings suggest that the nutritional intervention, particularly the refeeding protocol, was effective in preventing refeeding hypophosphatemia (RH) in critical patients with AN and partially promoting physical recovery in this population.

### 3.2. Pearson’s Correlation Matrix Analysis

To identify potential connections between the evaluated parameters, Pearson’s correlation analysis was conducted by selecting the most significant variables identified by the paired *t*-test and VIP scores from PLS-DA.

As depicted in Table 2, weight showed a moderate positive association with AC (0.414, *p* < 0.05). According to BIVA, weight exhibited moderate associations with FM (0.424, *p* < 0.001), TBW (0.351, *p* < 0.05), and ECW (0.318, *p* < 0.05). Interestingly, a very strong correlation was observed with FFM (0.805, *p* < 0.001) and BCM (0.789, *p* < 0.05), suggesting the very lean status of AN patients upon admission, as expected. In terms of HGS values, weight demonstrated a moderate association (0.510, *p* < 0.001), implying that patients with a higher body weight may exhibit improved HGS measures. This finding is consistent with a recent review demonstrating significant muscle atrophy and functional loss in AN patients [44]. These findings suggest that improving HGS could be an important outcome to target during the process of weight regain.

The most significant evidence from the biochemical analysis revealed a negative-to-moderate correlation between weight and CRP (−0.373, *p* < 0.05), indicating that extremely low body weight is associated with higher serum CRP levels. In terms of BMI correlation, the most notable association was observed with FM (0.593, *p* < 0.001) and NU parameters. Interestingly, BMI exhibited a strong negative association with the RF-Y axis (−0.828, *p* < 0.05) and T-SAT (−0.842, *p* < 0.05). These findings suggest that upon admission, AN patients may not only present with reduced total muscle mass but also significant depletion of lower limb muscle and central adiposity, as previously reported [30,31]. Therefore, focusing on muscle regain must play a central role in the physical recovery of these patients.

According to BIVA-related parameters, the most significant association observed was the negative correlation between FM and the duration of hospital admission (−0.347, *p* < 0.05), suggesting that patients with higher body fat mass may have a shorter hospital stay. This result is extremely interesting as the early detection of eating disorders could lead to less severe physical consequences such as central adiposity depletion, resulting in shorter hospital stays and reduced healthcare costs. To the best of our knowledge, this is the first study to correlate body composition with the duration of hospital admission. These findings are particularly noteworthy as they allow for a focus not only on weight but also on body fat mass as a potential target for monitoring physical recovery. Additionally, FM exhibited a moderate association with calcium levels (0.331, *p* < 0.05), which may be related to bone metabolism. While this study did not investigate bone mineral density (BMD), it has been noted that gaining FM has been identified as a crucial element linked with BMD enhancement in individuals with AN [45,46]. Achamrah et al. underscored that attaining normal bone levels is not solely tied to weight gain, emphasizing the significance of acknowledging the contribution of fat mass to the underlying mechanisms of osteoporosis and osteopenia in AN [45].

On the other hand, BCM values were closely associated with weight (0.789, *p* < 0.001) and BMI (0.400, *p* < 0.001). Moreover, BCM exhibited a strong association with hand strength (0.671, *p* < 0.001), suggesting that active cellular mass contributes to strength in these patients. Similarly, an increase in BCM was negatively associated with higher markers of inflammation such as CRP (−0.460, *p* < 0.001). Furthermore, HGS was associated with most muscle-related BIVA parameters such as TMM (0.473, *p* < 0.001), ASMM (0.482, *p* < 0.001), and FFM (0.658, *p* < 0.001). These findings underscore the validity of BIVA and HGS for analyzing muscle status and its functionality. Likewise, HGS may serve as a valuable and efficient predictor during consultations to assess muscle deterioration.

Regarding muscle-related NU parameters, RF-CSA showed strong associations with FFM (0.883, *p* < 0.05), TMM (0.966, *p* < 0.001), and ASMM (0.965, *p* < 0.001). These results also confirm the validity of this method for analyzing body composition and its reliability compared to other methods such as BIVA. Importantly, RF-CSA was not associated with body weight, BMI, or hydration status, indicating that it could be a useful tool when certain AN-related behaviors occur, such as vomiting or purgative use, which can compromise body weight and its derived measures.

Finally, T-SAT exhibited a strong and negative association with BMI (−0.842, *p* < 0.05) and AC (−0.937, *p* < 0.001) values upon admission. Once more, NU appears to be a valuable tool for monitoring physical recovery in AN patients, indicating a significant reduction in body fat composition upon admission. Similarly, patients with higher central adiposity were associated with longer RF (0.933, *p* < 0.001), as demonstrated in Table 2. These findings are supported by a recent review that highlighted significant muscle atrophy during weight loss-related starvation in AN patients [44].

### 3.3. Partial Least Squares-Discriminant Analysis

PLS-DA stands out as a valuable algorithm used for both predictive and descriptive modeling, as well as for selecting discriminative variables. It has demonstrated notable efficacy in handling complex datasets across various fields, including public health [47]. In this research, a threshold higher than 1 (equivalent to *p* < 0.05) was set as the criterion for selecting the VIP score in PLS-DA, as shown by a dashed vertical line in Figure 2B. Similarly, Figure 1A illustrates the PLS-DA analysis focusing on patients’ admission and discharge from the EDHU. The analysis revealed a slight overlap of the groups. Notably, the time to discharge for critical AN patients was approximately 50 days. However, there was an interesting trend in the spatial distribution towards the top-right part of the plot, indicating a tendency to differentiate between patients at admission and discharge times. The VIP score indicated that over 34% of the studied parameters might contribute to the observed effects. Notably, HGS values were identified by VIP score as the most significant variable for distinguishing between EDHU admission and discharge. As mentioned earlier, this parameter was also related to most of the BIVA muscular-related parameters. Similarly, body composition parameters such as FM, FFM, and ECW were also identified by VIP score as modulators of physical recovery, corroborated by Pearson’s correlation analysis. Additionally, the reduction of CRP after the intervention was also identified as a contributor to these differences.

Interestingly, among the 11 parameters identified by VIP score as the main contributors before and after the intervention, 27% of the parameters were BIVA-related, 27% were from classical anthropometry, 27% were from biochemical analysis, and 18% were from HGS and NU. These findings are highly significant as they suggest that the rapid implementation of these nutritional assessment methods could enhance the screening for physical recovery in AN patients, thereby facilitating the provision of personalized therapeutic interventions.

## 4. Conclusions

In conclusion, this comprehensive study sheds light on the intricate interplay between various physiological parameters during the multidisciplinary intervention for patients with AN. Numerous BIVA-related parameters such as fat and free fat mass were partially restored. Similarly, NU showed results in assessing body composition changes such as total abdominal fat tissue, correlating with clinical outcomes such as free fat mass and appendicular muscle mass. Hand dynamometry reached the normality percentile, demonstrating a significant association between grip strength and body composition parameters such as free fat mass and appendicular muscle mass. Leveraging advanced techniques such as BIVA, NU, HGS, and biochemical analysis alongside classical anthropometry, the research reveals nuanced insights into the physical recovery process. Notably, the significant associations identified between body composition parameters, inflammatory markers, and functional indicators underscore the complexity of AN management and highlight the potential for personalized therapeutic approaches tailored to individual patient needs.

Furthermore, the integration of predictive modeling techniques such as PLS-DA offers valuable insights into the key contributors to physical recovery before and after intervention. With hand strength, BIVA-related parameters (extracellular water and fat and free fat mass), classical anthropometry (weight, BMI, and AC), biochemical markers (glucose, CPR, and phosphorus), and NU (RF-*X*-axis) emerging as significant predictors, the study emphasizes the importance of a multidimensional approach in monitoring and evaluating AN patients.

The limitation of the present research lies in the nature of the techniques employed. Although BIVA and NU serve as good alternatives to DEXA, they necessitate an initial outlay to acquire the equipment. Similarly, both methods rely on trained personnel, and interoperator variability may exist. Furthermore, while multivariate analysis can identify important variables, it does not establish causal associations. However, the utilization of both VIP scores and Pearson correlation enables the assignment of a specific marker’s role in the observed effect.

These findings highlighted the potential for the rapid implementation of advanced nutritional assessment methods to enhance screening and optimize therapeutic strategies, ultimately improving outcomes for individuals undergoing treatment for anorexia nervosa.

## Figures and Tables

**Figure 1 nutrients-16-01539-f001:**
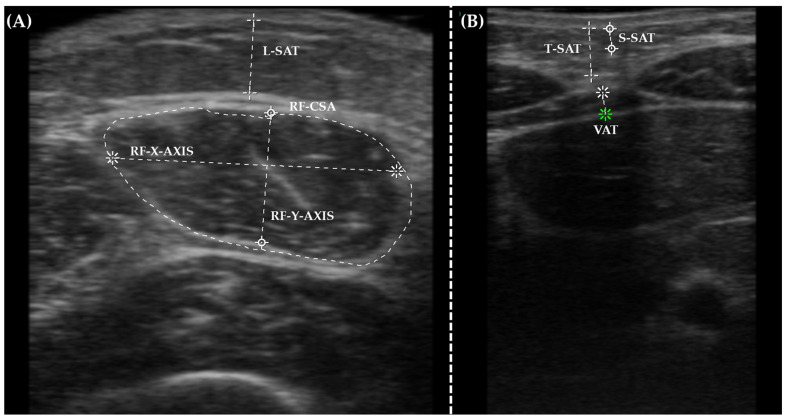
Illustrative captures of Nutritional Ultrasound^®^ in the (**A**) rectus femoris and (**B**) abdomen.

**Figure 2 nutrients-16-01539-f002:**
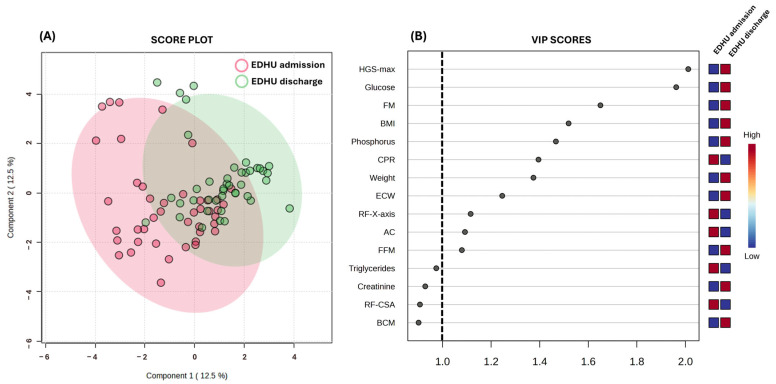
(**A**) PLS-DA score plot obtained from the mean values of the 32 parameters evaluated. (**B**) VIP score plots for the top 15 most important features by PLS-DA. The heatmap indicates the relative importance of the specific parameter in the different subpopulations and the dashed line represents statistical significance at *p* < 0.05. Abbreviations: AC: arm circumference; BCM: body cell mass; CPR: C-reactive protein concentration; ECW: extracellular water volume; FFM: fat-free mass; FM: fat mass; HGS max: maximum handgrip strength; RF-*X*-axis: rectus femoris *X*-axis length.

**Table 1 nutrients-16-01539-t001:** Population characteristics before and after psychiatric and nutritional intervention.

	**EDHU Admission**	**95% CI Admission**	**EDHU Discharge**	**95% CI Discharge**	** *p* ** **-Value**
**Anthropometry**					
Weight (kg)	37.4 (4.5)	28–47.1	42.8 (4.5)	32.1–51.9	<0.001
BMI (kg/m^2^)	14.3 (1.5)	11.4–17.3	16.3 (1.1)	13.1–18.6	<0.001
AC (cm)	18.2 (6.1)	14.18–21.5	19.9 (9.2)	16–23	<0.001
CC (cm)	27.8 (11.2)	20–31	29.1 (14.3)	23–33	0.052
TST (mm)	4.4 (2.7)	1.2–8.5	5.7 (3.3)	1.5–11	<0.01
**BIVA**					
PhA (°)	4.8 (0.7)	3.3–6	4.7 (0.5)	3.5–5.9	0.972
FM (kg)	3.5 (2)	1.7–10	5.3 (2.7)	1.7–12.6	<0.001
FFM (kg)	33.9 (3.8)	26.2–42.6	37.5 (4.1)	25.8–47.9	<0.001
TBW (L)	26.1 (2.7)	19.6–31.9	29.2 (7.3)	21.4–71.6	<0.05
ECW (L)	13.3 (2.0)	9.6–18	15 (4.4)	8.9–38.4	<0.05
BCM (kg)	15.8 (2.5)	10.2–20.9	17.4 (2.2)	11.3–21.8	<0.01
TMM (kg)	18.1 (2.5)	13.3–23	19.2 (2.8)	14.2–27.6	0.110
ASMM (kg)	12.5 (1.7)	9–16.2	13.4 (1.9)	10–18	<0.05
**Functional measurement**					
HGS max (kg)	21.6 (9.1)	8–35	25.9 (12.3)	14–37	<0.05
**Nutritional Ultrasound^®^**					
RF-CSA (cm^2^)	3.2 (1.5)	1.4–4.0	3.7 (1.3)	3–4.4	0.284
RF-*X*-axis (cm)	3.4 (1.6)	2.9–4.0	3.2 (1.1)	2.9–3.4	0.750
RF-*Y*-axis (cm)	1.2 (0.55)	0.8–1.7	1.5 (0.5)	1.3–1.7	0.413
L-SAT (cm)	0.4 (0.26)	0.1–1.1	0.7 (0.27)	0.4–1.3	0.270
T-SAT (cm)	0.5 (0.3)	0.2–1.1	0.9 (0.3)	0.7–1.3	<0.05
S-SAT (cm)	0.3 (0.2)	0.1–0.8	0.4 (0.2)	0.3–0.6	0.074
VAT (cm)	0.3 (0.1)	0.1–0.5	0.4 (0.1)	0.2–0.6	0.154
**Biochemical analysis**					
Glucose (mg/dL)	69.4 (27.5)	41–85	76.7 (14.5)	56–93	<0.01
Creatinine (mg/dL)	0.7 (0.3)	0.5–1.1	0.6 (0.2)	0.5–0.9	0.051
Proteins (g/dL)	6.6 (2.7)	4.5–8.5	6.8 (2.2)	4–8.4	0.541
Albumin (mg/dL)	4.3 (1.7)	3.1–5.7	4.4 (1.2)	3.3–5.7	0.731
Prealbumin (mg/dL)	29.8 (21.0)	16–101	27.3 (7.2)	20–37	0.488
CPR (mg/L)	4.3 (8.6)	0.2–32	0.5 (0.5)	0.2–3.2	<0.05
Total cholesterol (mg/dL)	160.9 (98.4)	0.2–403	174.1 (60.2)	1.5–269	0.373
Triglycerides (mg/dL)	96.6 (59)	30–280	70 (41.5)	21–205	<0.05
Calcium (mg/dL)	9 (3.4)	7.8–10.4	9.1 (2.4)	7.7–9.9	0.455
Phosphorus (mg/dL)	3.6 (1.6)	1.9–5.3	4.3 (0.8)	3.5–5.2	<0.001
Magnesium (mg/dL)	2.0 (0.8)	1.8–2.9	1.9 (0.5)	1.6–2.3	0.099
Potassium (mg/dL)	4.2 (1.7)	3.6–5	4.2 (0.8)	1.4–5.3	0.790

Data are expressed as mean ± standard deviations. Abbreviations: AC: arm circumference; ASMM: appendicular skeletal muscle mass; BCM: body cell mass; CC: calf circumference; CI: confidence interval; CPR: C-reactive protein concentration; ECW: extracellular water volume; FFM: fat-free mass; FM: fat mass; HGS max: maximum handgrip strength; PhA: phase angle; RF-CSA: Rectus femoris cross-sectional area; RF-*X*-axis: rectus femoris *X*-axis length; RF-*Y*-axis: rectus femoris *Y*-axis length; SAT: Subcutaneous adipose fat of leg (L), and superficial (S) and total (T) abdominal; TST: triceps skinfold thickness; TBW: total body water volume; TMM: total muscle mass; VAT: visceral adipose tissue.

**Table 2 nutrients-16-01539-t002:** Pearson’s correlation analysis of classical anthropometry, BIVA, handgrip strength, and nutritional ultrasound^®^ with the rest of the parameters on admission time.

	**Anthropometry**		**BIVA**		**Functional**	**Ultrasound**	
	**Weight**	**BMI**	**FM**	**BCM**	**HGS Max**	**RF-CSA**	**T-SAT**
**Anthropometry**							
Weight (kg)	1 **	0.510 **	0.424 **	0.789 **	0.386 *	-	-
BMI (kg/m^2^)	0.510 **	1 **	0.593 **	0.400 **	-	-	−0.842 *
AC (cm)	0.414 *	0.516 **	0.430 *	-	-	-	−0.937 **
CC (cm)	-	-	-	-	-	-	-
TST (mm)	-	0.430 *	0.589 **	-	−0.414 *	-	-
**BIVA**							
PhA (°)	-	0.491 **	0.424 **	0.554 **	-	-	-
FM (kg)	0.424 **	0.593 **	1 **	-	−0.372 *	-	-
FFM (kg)	0.805 **	-	-	0.818 **	0.658 **	0.883 *	-
TBW (L)	0.351 *	-	-	0.365 **	0.359 *	-	-
ECW (L)	0.318 *	-	−0.365 *	-	0.399 *	-	-
BCM (kg)	0.789 *	0.400 **	-	1 **	0.671 **	-	-
TMM (kg)	-	-	−0.435 **	0.356 **	0.473 **	0.966 **	-
ASMM (kg)	-	-	−0.406 **	0.387 **	0.482 **	0.965 **	-
**Functional parameters**							
HGS max (kg)	0.386 *	-	−0.372 *	0.671 **	1 **	-	-
Time spent on the unit (days)	-	-	−0.347 *	-	-	-	-
**Nutritional Ultrasound^®^**							
RF-CSA (cm^2^)	-	-	-	-	-	1 **	-
RF-*X*-axis (cm)	-	-	-	-	-	-	-
RF-*Y*-axis (cm)	-	−0.828 *	-	-	-	-	0.933 **
L-SAT (cm)	-	-	-	-	-	-	-
T-SAT (cm)	-	−0.842 *	-	-	-	-	1 **
S-SAT (cm)	-	-	-	-	-	-	-
VAT (cm)	-	-	-	-	-	-	-
**Biochemical analysis**							
Glucose (mg/dL)	-	-	-	-	-	-	-
Creatinine (mg/dL)	-	-	-	-	-	-	0.918 **
Proteins (g/dL)	-	-	0.364 *	-	-	-	-
Albumin (mg/dL)	−0.351 *	-	-	−0.324 *	-	-	-
Prealbumin (mg/dL)	-	-	-	-	-	-	-
CPR (mg/L)	−0.424 **	−0.373 *	-	−0.460 **	-	-	-
Total cholesterol (mg/dL)	-	-	-	-	-	-	-
Triglycerides (mg/dL)	−0.369 *	−0.352 *	-	−0.365 **	-	-	-
Calcium (mg/dL)	-	-	0.331 *	-	-	-	-
Phosphorus (mg/dL)	-	-	-	-	-	-	-
Magnesium (mg/dL)	0.342 *	-	-	-	-	-	-
Potassium (mg/dL)	-	-	-	-	-	-	-

Pearson’s Correlation Analysis was assumed significant with *p*-value < 0.05 (* means < 0.05 and ** means *p* < 0.001); “-” means absence of association. Abbreviations: AC: arm circumference; ASMM: appendicular skeletal muscle mass; BCM: body cell mass; CC: calf circumference; CPR: C-reactive protein concentration; ECW: extracellular water volume; FFM: fat-free mass; FM: fat mass; HGS max: maximum handgrip strength; PhA: phase angle; RF-CSA: Rectus femoris cross-sectional area; RF-*X*-axis: rectus femoris *X*-axis length; RF-*Y*-axis: rectus femoris *Y*-axis length; SAT: Subcutaneous adipose fat of leg (L), and superficial (S) and total (T) abdominal; TST: triceps skinfold thickness; TBW: total body water volume; TMM: total muscle mass; VAT: visceral adipose tissue.

## Data Availability

The original contributions presented in the study are included in the article, further inquiries can be directed to the corresponding authors.

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
