# Peer review of "Bioelectrical Impedance Vector Analysis, Nutritional Ultrasound^®^, and Handgrip Strength as Innovative Methods for Monitoring Critical Anorexia Nervosa Physical Recovery: A Pilot Study"

_nutrients, 2024, doi:10.3390/nu16101539_

Round 1
Reviewer 1 Report
Comments and Suggestions for Authors
The Authors decided to assess numerous nutritional parameters in anorexia nervosa patients. The study is interesting, however I must express my concerns about the submitted manuscript.
Please provide basic statistical data in the abstract to support the presented conclusions.
The introduction is very short and does not provide enough insight into the study rationale. Please elaborate on each topic in the introduction.
The is only one sentence about ED, and then the rest of the introduction refers to AN. I suggest updating the manuscript title, as it focused only on one of the possible EDs, so the title is misleading.
Please underline the strong points and novelty of the presented study, what was the knowledge gap (the are available studies that have already used these techniques), and what was it aimed to prove? What was the main hypothesis? The number of the analyzed parameters is so large that it is hard for the reader to easily find the main aim of this study.
Please provide also limitations of the study in a dedicated paragraph.
Line 77 – were all the patients adults?
Line 80 – what were the inclusion criteria? Why 2 patients with unspecified EDs were included in the study?
Line 120 – please provide details of the BIA measurement procedure. The cited reference is not the AN study, so please describe the procedure in the presented manuscript.
Line 134 – please provide details of the ultrasound measurement procedure.
Why other anthropometrical measurements such as circumferences were not included?
Line 155 – was the data distribution normal?
What was the period between admission and discharge? Was it the same for all of the patients? If not, what was the base for the decision of the discharge?
Line 162 – Please separate the results and the discussion paragraphs according to the journal guidelines
Line 185 – there are more available papers focused on phase angle in AN patients - please discuss them
There is available the study regarding the BIVA application is AN (10.1002/erv.1166) please discuss it.
Line 225 – The sentence need corrections. I assume that the information about healthy controls refers to other studies (please correct it to make it more clear). If not, so how were the healthy controls recruited? There is no information about it in the methods section.
Line 199 - There is available a study that used ultrasound to assess fat mass (10.1016/j.clnu.2018.12.031)
Line 232 – there are available studies upon this topic, this is not the first one. Please check 10.1016/j.nut.2020.111133 10.1080/15622975.2020.1774652 10.1002/erv.2839
Line 300 (table) – why the is no information about phase angle and HGS or ultrasound parameters correlation?
Conclusions do not show the main findings of the study and do not refer to the stated hypothesis. What is the value of the performed analyzes?
Line 406 – please provide the information about the ethical board endorsement with the decision number.
Comments on the Quality of English Languageminor corrections needed
Author Response
The authors express sincere gratitude to the reviewers for dedicating their time to undoubtedly improving the quality of this manuscript. Please see the attachment.

Reviewer 2 Report
Comments and Suggestions for Authors
My feedback:
Abstract:
· The abstract summarizes the study's aim, methods, key findings, and implications, effectively preparing the reader for a detailed exploration of the main text. However, the sentence structure and flow could be improved for better readability. For example, the phrase "After the intervention, numerous BIVA-related parameters such as fat and free fat mass were partially restored," could be made more concise and impactful.
· The mention of "43 female ED patients" is good as it specifies the study population. However, it would be helpful to briefly note the type of eating disorders included if the study was not limited to a single type. Clarifying whether these were cases of anorexia nervosa, bulimia, or other eating disorders would provide a better context for the reader.
· The results, which emphasize gains in grip strength and body composition, are presented with a fair amount of detail. To support the integrity of the published data, the abstract can, however, briefly discuss statistical significance—or lack thereof. It would be more substantive, for instance, to state that when appropriate, changes attained statistical significance.
· The abstract uses acronyms such as DEXA, BIVA, and others without defining them on first use within the abstract. While these may be familiar to specialists, defining them at first mention would enhance readability and accessibility for a broader audience. For example, "Dual-Energy X-Ray Absorptiometry (DEXA)."
1. Introduction
· The introduction skillfully draws attention to the incidence and consequences of AN in teenagers by citing relevant data and the COVID-19 pandemic's escalation of these tendencies. The references in the literature appear to be from 2015 and earlier, so it's possible that more recent research or statistics will provide us a more up-to-date picture of the state of the condition.
· The authors mention the limitations of DEXA and propose BIVA, NU, and HGS as alternative methods due to their affordability, portability, and non-invasiveness. While the advantages of these methods are listed, the introduction lacks a critical comparison with existing methods beyond DEXA, such as bioelectrical impedance analysis (BIA) or MRI, which might also offer similar benefits. A more detailed justification for why these three specific methods were chosen over others would strengthen the rationale.
· The introduction tends to generalize the impact of AN without differentiating between the various forms of EDs that might respond differently to the tested methods. It would be beneficial to clarify whether the methods proposed have specific relevance to AN or if they are also applicable to other forms of EDs like bulimia nervosa or binge eating disorder.
· The study's objectives are implied rather than explicitly stated. For clarity, directly stating the research hypothesis and objectives at the end of the introduction could help frame the research questions more definitively. This would guide the reader to what specifically the study aims to address beyond the broad goal of "monitoring physical recovery."
2. Materials and Methods
· The selection criteria for patients should be detailed further. Were there specific criteria related to the severity of the eating disorder, comorbid conditions, or previous treatment histories that defined eligibility? This is crucial for understanding the generalizability of the study findings.
2.2. Psychiatric and Nutritional Intervention
· This subsection would benefit from more detailed descriptions of the interventions used. For instance, what specific therapeutic approaches were employed in the therapeutic dining room? What were the qualifications of the staff involved?
· How were the interventions personalized for each patient? Details on how the interventions were adapted based on individual patient needs, or responses would provide deeper insight into the treatment methodology.
2.3. Anthropometric Measurements
· It is good that standardized procedures were followed for anthropometric measurements; however, the reliability and calibration of the equipment (e.g., stadiometer and weighing scale) should be mentioned. Also, who performed these measurements, and what training did they have?
2.4. Bioelectrical Impedance Vector Analysis
· The use of a 50 kHz phase-sensitive impedance analyzer is well justified. Nevertheless, including information on how patients were prepared for the measurement (e.g., hydration status, physical activity restrictions) would help replicate the study.
2.5. Nutritional Ultrasound
· The methodological description is precise, which is excellent. However, clarifying why the rectus femoris and abdominal measurements were chosen and their relevance to the study outcomes would strengthen this section.
· How was inter-operator variability addressed in ultrasound measurements, given that multiple professionals were involved?
2.6. Handgrip Strength Analysis
· The methodology for measuring handgrip strength is well-described. Consider specifying whether adjustments or calibrations were made based on patient age, height, or gender, as these can influence handgrip strength.
3. Results & Discussion
· The manuscript presents a comprehensive set of data, including standard deviations and p-values, which is commendable as it indicates variability and statistical significance. Nonetheless, including confidence intervals, particularly for key measurements such as body weight, BMI, and handgrip strength, could further aid in interpreting the clinical significance of the findings.
· The use of Pearson's correlation matrix is robust; however, the explanation of these correlations' implications in the clinical context of eating disorders could be expanded. Specifically, discussing how these correlations can influence treatment strategies would be valuable.
2. Interpretation of BIVA Data:
· The discussion on the stability of PhA values despite nutritional intervention is intriguing, but it somewhat contradicts findings from other studies mentioned. It would be beneficial to explore potential reasons for this discrepancy in more detail, possibly considering the severity of malnutrition in the study cohort as a contributing factor.
· The post-intervention rise in FM and FFM is well addressed. However, the conversation would be more applicable to therapeutic practice if it made a connection between these modifications and possible clinical outcomes or the healing process in eating disorder patients.
3. Nutritional Ultrasound and Handgrip Strength Analysis:
· In this context, nutritional ultrasound is new, but there is a need for more discussion of its findings, especially the non-significant changes in muscle-related metrics. The usefulness of this approach might be validated and a wider perspective could be provided by a comparison with other research that used comparable methodologies.
· The study highlights the increase in handgrip strength and its implications for patient functionality and recovery. Potential rehabilitation programs or interventions could be discussed here to translate these findings into practical applications.
4. Biochemical Analysis:
· The discussion on normalizing biochemical markers such as glucose and the non-significant changes in serum protein markers is well-handled. Yet, despite nutritional interventions, the clinical implications of these stable protein levels warrant further exploration, particularly in terms of patient outcomes and recovery metrics.
· The reduction in CRP levels post-intervention is discussed with reference to inflammation; however, expanding on how this might affect the overall health and recovery trajectory of patients would strengthen the discussion.
Comments on the Quality of English LanguageThe article's abstract and body demonstrate how to articulate concepts and present the findings organizationally and clearly. For more professionalism and clarity, a few minor grammar faults and occasionally difficult phrasings might be fixed. The readability and general presentation of the study findings would be improved by changing the sentence structure, maintaining consistency in the verb tenses, and eliminating superfluous words.
Author Response

(The authors gave the same response as above.)
